# Sexual Difference in the Optimum Environmental Conditions for Growth and Maturation of the Brown Alga *Undaria pinnatifida* in the Gametophyte Stage

**DOI:** 10.3390/genes11080944

**Published:** 2020-08-16

**Authors:** Yoichi Sato, Hikaru Endo, Hiroki Oikawa, Koichi Kanematsu, Hiroyuki Naka, Miho Mogamiya, Shigeyuki Kawano, Yusuke Kazama

**Affiliations:** 1Bio-resources Business Development Division, Riken Food Co., Ltd., Miyagi 985-0844, Japan; mih_mogamiya@rikenfood.co.jp; 2Nishina Center for Accelerator-Based Science, RIKEN, Saitama 351-0198, Japan; 3Faculty of Fisheries, Kagoshima University, Kagoshima 890-0056, Japan; h-endo@fish.kagoshima-u.ac.jp; 4Connected Solutions Company, Panasonic Co., Tokyo 104-0061, Japan; oikawa.hiroki@jp.panasonic.com (H.O.); kanematsu.k@sim24.co.jp (K.K.); naka.hiroyuki@sim24.co.jp (H.N.); 5SiM24 Co., Ltd., Osaka 540-6104, Japan; 6Future Center Initiative, The University of Tokyo, Saitama 277-0871, Japan; kawano@edu.k.u-tokyo.ac.jp; 7Faculty of Bioscience and Biotechnology, Fukui Prefectural University, Fukui 910-1195, Japan

**Keywords:** sexual difference, *Undaria pinnatifida*, gametophyte, optimum conditions, growth, maturation, fertilization, response surface methodology

## Abstract

*Undaria pinnatifida* is an annual brown kelp growing naturally in coastal areas as a major primary producer in temperate regions and is cultivated on an industrial scale. Kelps have a heteromorphic life cycle characterized by a macroscopic sporophyte and microscopic sexual gametophytes. The sex-dependent effects of different environmental factors on the growth and maturation characteristics of the gametophyte stage were investigated using response surface methodology. Gametophytes were taken from three sites in Japan: Iwate Prefecture, Tokushima Prefecture, and Kagoshima Prefecture in order to confirm the sexual differences in three independent lines. Optimum temperature and light intensity were higher for males (20.7–20.9 °C and 28.6–33.7 µmol m^−2^ s^−1^, respectively) than females (16.5–19.8 °C and 26.9–32.5 µmol m^−2^ s^−1^), and maturity progressed more quickly in males than females. Optimum wavelengths of light for growth and maturation of the gametophytes were observed for both blue (400–500 nm, λ_max_ 453 nm) and green (500–600 nm; λ_max_ 525 nm) lights and were sex-independent. These characteristics were consistent among the three regional lines. Slower growth optima and progress of maturation could be important for female gametophytes to restrict fertilization and sporophyte germination to the lower water temperatures of autumn and winter, and suggest that the female gametophyte may be more sensitive to temperature than the male. The sexual differences in sensitivity to environmental factors improved the synchronicity of sporeling production.

## 1. Introduction

Sexual dimorphism in which males and females differ physiologically or morphologically is common among eukaryotes. Theory indicates that the evolution of sexual dimorphism is facilitated by the XY (or ZW) sex-determination system [1]. Many algal species exhibit sexual dimorphism with a broad range of diversity in the difference of size between male and female gametophytes (see review [2,3]). Sex in these algae is determined by the UV system in which sexuality is displayed at the haploid stage of their life cycle. Because both the U and V chromosomes are directly exposed to purifying selection during a longer gametophyte period than that present in XY or ZW systems, sexual dimorphisms in physiological characteristics also may have evolved in the algae. In brown algae, several sexually dimorphic traits in relation to physiology have been suggested, including pheromone release and sensitivity to temperature (see review in [4]). Although it may be considered that males and females respond differently to biotic factors, this is not supported by the limited data currently available. Therefore, to confirm whether such effects exist requires further accumulation of data concerning sexual differences in the response to biotic factors.

*Undaria pinnatifida* (Harvey) Suringar (Phaeophyceae, Laminariales; Japanese vernacular name ‘wakame’) is a large annual brown alga growing naturally in temperate coastal areas. It is a major commercial species which has been cultivated in Japan, Korea, and China since the 1950s as an edible seaweed, and there is now a large commercial market [5]. It has a heteromorphic life cycle characterized by macroscopic sporophyte and microscopic gametophyte stages (Appendix Aa) [6]. The visible sporophyte grows from winter to spring and its morphological features include blade, stipe, sporophylls, and holdfast (Appendix Ab–d). The sporophylls are formed along the lower part of the stipe in spring from which they release millions of asexual spores. The spores attach to substrates such as rocks and germinate into dioicious gametophytes. 

At the gametophyte stage, the male and female plants grow separately during summer. From autumn to winter, when the seawater temperature drops, the gametophytes produce sporophytes through sexual reproduction. Considering the optimization of fertilization opportunities, male gametophytes of *U. pinnatifida* might be expected to grow faster and mature sooner than female gametophytes during periods when the seawater temperature is decreasing. In another kelp, *Laminaria digitata,* it has been shown that the reproductive period of males is longer than that of females [7]. With female gametophytes limited to a short reproductive period, it might be expected that it is important to ensure that male gametes are available to fertilize them to successfully form sporophytes. This could be achieved by male gametophytes having a longer reproductive period than females to ensure overlap with the shorter female reproductive period. Therefore, the working hypothesis is that the optimum condition for growth and maturation of gametophytes will differ between males and females. 

During cultivation of laminarian kelps, manipulation of the life cycle transition between gametophytes and sporophytes is common practice to enable year-round production of sporophytes as sporelings [8,9]. The microscopic gametophytes are probably a factor in the wide distribution of this species worldwide since the 1970s, as a result of international shipping and mariculture. Currently, *U*. *pinnatifida* has invaded many coastal regions worldwide, including the south-western Atlantic [10,11], north-eastern Atlantic [12], north-eastern Pacific [13], the Mediterranean Sea [14], and waters off Australia [15] and New Zealand [16]. The rapid expansion of its distribution in such a short time has disturbed the invaded ecosystems, and it is noted as one of the world’s 100 worst invasive alien species [17]. To control the propagation and invasion of *U. pinnatifida*, better knowledge is required of the biology of its microscopic gametophyte stage. In addition, any prediction of the future impacts of global warming on kelps must consider the responses of both the sporophyte and gametophyte stages [18,19,20]. 

The optimal environmental conditions for growth and maturation of the *U. pinnatifida* gametophyte have been well studied and include a seawater temperature of 19–25 °C, light intensity of 35–100 µmol m^−2^ s^−1^ and a wavelength within the range of 460–560 nm [21,22,23,24,25,26,27]. All of these experiments were performed mostly with a mixture of male and female gametophytes. However, considering the life cycle of *U. pinnatifida* (as noted above), the growth and maturation characteristics of males and females should be studied separately. Furthermore, previous studies have examined the effects of several environmental factors separately on the responses in terms of growth and maturation but have not considered the effects of various environmental factors combined. In order to analyze the growth and maturation conditions of *U. pinnatifida* at the gametophyte stage according to sex, it is necessary to investigate the interaction of the effects of several environmental factors in combination. Including different regional lines of *U. pinnatifida* allows exclusion of the effects of non-sexual genetic diversity on the optimum growth conditions. The lines we have used here are classified into two groups, which are known to show ecotypic differences at the sporophyte stage (photosynthetic rates and carbon and nitrogen assimilation [28]; nutrient-uptake kinetics [29]; and morphological characteristics [30]) but the gametophyte stage has not been examined previously. 

Response surface methodology (RSM) is a useful statistical method of analysis to find the optimum conditions among multiple factors using minimal measurement data [31,32]. It has been used often in other fields, such as in evaluating optimum conditions for fungi culture [33,34]; chemical synthesis [35]; food cultivation and processing [36,37]; and upregulation of oil and other useful components in micro-organism cultures [38,39]. RSM consists of three steps. (1) Experimental conditions are decided according to initially designed experiments to obtain data efficiently for building the RSM model. (2) Culture experiments are carried out under reproducible conditions and the experimental results are fitted to the RSM model. For modeling, a first-order polynomial (including interaction between factors) is often used. (3) A search is performed to look for optimum values among parameters correlated from the RSM model, using mathematical optimization methods [40,41]. 

The aim of the present study was to reveal any sexual differences in optimum growth and maturation conditions of *U. pinnatifida* gametophytes in three different regional populations around the coastal waters of Japan. Gametophytes were collected from Iwate Prefecture (Pref.) in north eastern Honshu, Tokushima Pref. in Shikoku, and Kagoshima Pref. in Kyushu (Figure 1). RSM was used to analyze the results from suitably designed experiments. 

## 2. Materials and Methods

### 2.1. Preparation and Culture of Gametophytes

One individual mature sporophyte of *Undaria pinnatifida* was collected at each of three sites (Figure 1): Ohno Bay, Iwate Pref. (IWT; 38°97′17′’ N, 141°72′12′’ E), in June, 2015; the coast of Naruto City, Tokushima Pref. (TKS; 34°22′48′’ N, 134°64′22′’ E), in April, 2016; and Kagoshima Bay, Kagoshima Pref. (KGS; 31°59′99′’ N, 130°56′76′’ E), in April, 2017. Blade sections (2 cm × 2 cm) were excised from sporophylls and zoospores were released into sterilized seawater [29,30]. Zoospores were used to inoculate a 9-cm dish containing 30 mL of PESI medium [42] at a concentration low enough to allow individuals to be distinguished clearly, and they were cultivated at 20 °C under white fluorescent light at an intensity of 50 µmol m^−2^ s^−1^. Gametophyte individuals were cultured separately in different microplate wells from early growth stages to those in which the sexes of the gametophytes are easily distinguished. After two weeks, filamentous gametophytes were segregated into males and females and cultivated separately. The largest male and female individuals were used for culture experiments.

Gametophytes of 0.2 g wet weight were broken up (using a PRO200 Homogenizer; Pro Scientific Inc., Oxford, CT, USA) at 5000 rpm for 2 min to obtain few-celled individual cells. Filaments composed of exactly five cells were selected and added to each well of a 24-well microplate containing 2 mL PESI medium [42]. Before starting this study, a number of replicates were tested (6, 12, and 24 replicates) and it was confirmed that a number of 24 replicates is sufficient for statistical analysis. All wells in each microplate each contained one gametophyte, with one microplate for each culture condition (growing and mature) for each sex and region. To provide independent culture conditions, 15 small incubators (CN–40A, Mitsubishi Electric Engineering Co., Ltd., Tokyo, Japan) and LED units (3LH–64, NK Systems Co., Ltd., Osaka, Japan) were prepared (Appendix A). To minimize fluctuations of the incubator temperature to ±0.5 °C (due to heat dissipation at the top of the LED unit), a stainless-steel plate was installed diagonally to prevent vertical stratification of the air inside. The LED light wavelengths used were blue (400–500 nm, λ_max_ 453 nm), green (500–600 nm; λ_max_ 525 nm), red (600–700 nm; λ_max_ 641 nm), and (by combination of these three wavelengths) white (Appendix A). Light intensity and wavelength were measured with a light analyzer (LA–105, NK Systems Co., Ltd. Osaka, Japan).

### 2.2. Measurement of Growth and Maturation of Gametophytes

Photomicrographs of all gametophytes were taken once every five days during the 25 days of cultivation. The area of each gametophyte was measured using a custom-made software, which automatically calculates the area of each gametophyte from images obtained by binarization processing. A representative image is shown in Appendix A. The relative growth rate (RGR) was calculated using the area, as in the following equation (Equation (1)): (1)$$\mathrm{RGR}\;\left({\mathrm{day}}^{-1}\right)=\ln(A_{t}/A_{0})/t$$

A_0_: Initial gametophyte area, A_t_: final gametophyte area after the experiment, t: number of days of cultivation. 

To evaluate the degree of maturation of each gametophyte (M_deg_), a five-level index (I–V) was devised based on the number of egg cells (females) or the ratio of branches with bushy tips forming spermatangia to the total number of branches (males). Representative photographs for each stage are shown in Figure 2. Observations were performed five times in total during 25 days of cultivation (at days 0, 6, 13, 19, and 25). 

### 2.3. Temperature Data

Daily temperature records over several years were obtained from each locality. These are summarized graphically in Appendix A after dividing monthly data into three bins (of 10, 10, and 8–11 days) and calculating the mean temperature for each bin. Temperature data for Iwate Pref. are from 2008 to 2017, as measured in Hirota Bay by the Iwate Fisheries Technology Centre. Data for Tokushima Pref. are from 2008 to 2017 measured in the sea off Naruto by the Forestry and Fisheries Technology Support Centre, Naruto Division. Data for Kagoshima Pref. are from 2009 to 2017, measured in Kagoshima Bay by staff at Kagoshima City Aquarium. 

### 2.4. Experimental Design 

A graphical tools approach to the experimental design [43] was used to define the experimental matrix in order to estimate the interacting effects of optimal conditions of temperature, light intensity, and wavelength on growth and maturation of gametophytes. The effects of two numeric factors (temperature, X_1_, and light intensity, X_2_) and one categorical factor (light wavelength, X_3_) were studied separately for each of two response variables. Response surface methodology (RSM) was used to estimate the effects of the independent variables (X_1_, X_2_, X_3_) on the gametophyte response variables growth (RGR, Y_1_) and maturity (M_deg_, Y_2_) in three regional lines. Fifteen different combinations of factors were investigated for gametophyte growth (Table 1) and 12 different combinations of factors were studied for gametophyte maturation (Table 2). The software package JMP (Version 11.2.1, SAS Institute Inc., Cary, NC, USA) was used to determine the experimental design matrix and perform the RSM statistical analysis.

### 2.5. Statistical Analysis

The response variables (Y) were related to the coded independent variables (Xi, i = 1, 2, and 3) by a second-order polynomial model using a least-square method (Equation (2)). The coefficients of the polynomial model were introduced by b_0_ (constant term), b_1_ and b_2_ (linear effects of seawater temperature and light intensity as main effects), b_3_ (interaction effect of seawater temperature and light intensity), and b_4_ and b_5_ (quadratic effects of seawater temperature and light intensity).
(2)$$Y_{1}={\;b}_{0}+\;b_{1}X_{1}+{\;b}_{2}X_{2}+{\;b}_{3}X_{1}X_{2}+{\;b}_{4}X_{1}^{2}+{\;b}_{5}X_{2}^{2}+{\;X}_{3}$$

Y_1_: response (RGR), X_1_: seawater temperature, X_2_: light intensity, X_3_: light color, b_0_: constant term, b_1_–b_5_: coefficients.

The response variables of maturation were related to the coded independent variables (Xi, i = 1, 2, and 3) by an ordered logistic regression (OLR) model (Equation (3)) because the degree of maturation data is not a continuous variable. Y_2_ was incorporated into OLR as a second-order polynomial model, with coefficients as for Eqation (2).
(3)$$P\left[n\right]=A\left[n\right]-A\left[n-1\right]=\;1/\left(1+\mathrm{Exp}\left(a\left[n\right]-Y_{2}\right)-1/\left(1+\mathrm{Exp}\left(a\left[n-1\right]-Y_{2}\right)\right)\right)$$


$Y_{2}={\;b}_{1}X_{1}+{\;b}_{2}X_{2}+{\;b}_{3}X_{1}X_{2}+{\;b}_{4}X_{1}^{2}+{\;b}_{5}X_{2}^{2}+{\;X}_{3}+{\;b}_{6}X_{4}$


P: Probability, A: cumulative probability, Y_2_: response (M_deg_), X_1_: seawater temperature, X_2_: light intensity, X_3_: light color, X_4_: cultivation days, b_0_: constant term, b_1_–b_6_: coefficients.

The RSM and OLR models were evaluated to determine a set of experimental conditions for the highest degree of maturation using a desirability function [40].

## 3. Results

### 3.1. Regional Differences of Undaria pinnatifida

Three regional lines with different morphological features at the sporophyte stage were used to investigate the sexual difference of gametophytes of *U. pinnatifida*. Particular differences in the sporophytes of these lines are visible for stipe length between the blades and sporophylls (Figure 1). According to information on the haplotype divergence of the mitochondrial loci of these three lines [44], the lines from the north-eastern Pacific coast (IWT and TKS) are classified within the same group, “northern Japan type”. The southern Japan area including KGS was classified into another group, the “Pacific central Japan type”. The distinguishing morphological features of KGS include a shorter total length and wider blade compared with IWT and TKS (Figure 1). Although the morphological features of IWT and TKS were similar at the sporophyte stage when they were cultivated under the same environmental conditions [30], they showed physiological differences in photosynthesis activity, and carbon and nitrogen assimilation [28]. It is therefore assumed that KGS, IWT, and TKS have different genetic backgrounds, and the morphological differences between IWT and TKS are emphasized by environmental factors. 

### 3.2. Growth and Maturation of Gametophytes

The results of gametophyte growth are presented in Table 1. In the cultivation assays, the RGR of male gametophytes varied significantly from 0.504 to 1.277 day^−1^ for IWT, from 0.246 to 1.743 day^−1^ for TKS, and from 0.454 to 1.975 day^−1^ for KGS. The RGR of the female gametophytes varied significantly from 0.128 to 1.134 day^−1^ for IWT, from 0.193 to 1.573 day^−1^ for TKS, and from 0.413 to 1.556 day^−1^ for KGS. At day 25, gametophytes grew to approximately 500 µm in diameter or 0.004–0.007 mg in wet weight. Representative photographs of female gametophytes derived from the three regions, cultivated under lights of one of three individual colors for 25 days, are shown in Figure 3. These color-related responses were common among the three regional lines. Gametophytes cultivated under red light showed little vegetative growth, whereas those cultivated under green light showed marked growth but without maturation. Gametophytes cultivated under blue light produced many eggs on females and spermatangia on males, with a high growth rate similar to those cultivated under green light. In particular, under blue light at 40 µmol m^−2^ s^−1^ and 20 °C (Assay 7 in Table 1), germination of sporophytes (see Figure 3, arrow, for example) was observed in 7 of the 24 female gametophyte individuals from IWT and 3 of the 24 from TKS, even though male and female gametophytes were incubated separately to avoid fertilization. 

The experimental results of male and female gametophyte maturation are presented in matrix form in Table 2 at day 25 (as a representative sample: data for days 0, 6, 13, and 19 are not shown). In the cultivation assay for maturation, full maturity (stage V) at the final date of cultivation varied from 0.01 to 0.76 for IWT, from 0.01 to 0.74 for TKS, and from 0.00 to 0.05 for KGS, depending on the cultivation conditions.

### 3.3. Optimum Conditions for Growth of Gametophytes 

The least-square method for each regional line for males and females was modeled by using RGR data of gametophytes using the experimental design of [43] (Appendix A). Summaries of the analysis of variance (ANOVA) for the selected predictive model is shown in Table 3. These analyses show that both males and females for all regional lines were statistically significant (*p* < 0.0001, Table 3). The calculated coefficients of all environmental factors are shown in Appendix A. Environmental factors significantly affected the RGR of gametophytes except for the following: seawater temperature (X_1_) of KGS males, and interaction effects between seawater temperature and light intensity (X_1_X_2_) of TKS males and KGS males and females. According to these models, the optimum conditions were achieved for the growth of male and female gametophytes at the maximum desirability of X_1_, X_2_, and X_3_ (see Appendix A, and presented graphically in Appendix A). Common to all three regional lines, the optimum seawater temperature for the growth of male gametophytes was higher than that for female gametophytes: the temperature ranges of males/females were, respectively, 20.7 °C/18.6 °C for IWT; 20.9 °C/16.5 °C for TKS; and 20.7 °C/19.8 °C for KGS. The optimum light intensity for growth of the male gametophytes was slightly higher than that of the female gametophytes, and was identical for all regional lines: intensities for males/females were, respectively, 33.7/32.5 µmol m^−2^ s^−1^ for IWT, 32.7/31.3 µmol m^−2^ s^−1^ for TKS, and 28.6/26.9 µmol m^−2^ s^−1^ for KGS. The optimum light color for growth of male and female gametophytes was blue or green, and growth under red was markedly inferior to that under blue, green, or white. Growth under white light was lower than under blue or green, but higher than under red. These responses to light color were common to all three regional lines. At the optimum conditions for all regional lines, the RGR values for males were higher than those of females (Figure 4). 

Comparing the three regional lines, although the optimum temperature for growth of males was almost the same (Table 4, 20.7–20.9 °C), females in KGS showed a higher growth range (Table 4, 16.5–19.8 °C) than the other regional line females. The optimal light intensity for KGS gametophytes was lower than that for IWT and TKS for both males and females (Table 4). There was a tendency for the coefficient values (b_2_) of X_2_^2^ to increase for both males and females in the order IWT < TKS < KGS: males/females 0.051/0.053 (IWT), 0.073/0.105 (TKS), and 0.112/0.111 (KGS) (Appendix A).

### 3.4. Optimum Conditions for Maturation of Gametophytes 

The optimum conditions for maturation were modeled by logistic regression analysis using the maturation stages of male and female gametophytes for each regional line [43] (Table 2). Summaries of likelihood ratio tests for the selected predictive model through all cultivation periods are shown in Table 5. These analyses show that models of both males and females for all regional lines were statistically significant (*p* < 0.001 in Table 5: the model equation is shown in Appendix A). The calculated coefficients of all factors are shown in Appendix A. For the male gametophytes, some factors did not significantly affect their maturation: X_1_ for IWT; X_2_ and X_2_^2^ for TKS; and X_3_, X_1_X_2_, and X_2_^2^ for KGS. For the female gametophytes, the interaction factor (X_1_X_2_) showed no significant effect on maturation for any of the regional lines. In the KGS line only, X_2_ had no significant effect on maturation. 

According to this model, the optimum conditions of seawater temperature and light intensity were obtained to achieve maturation stage V (Table 6). For IWT, the optimum temperature for the male gametophytes was 1.6 °C higher than that for the female gametophytes, and the optimum light intensity for the male gametophytes was lower than that for the female gametophytes (−11.2 µmol m^−2^ s^−1^). The optimum light wavelength for maturation to progress was blue for both male and female gametophytes. The differences between males and females regarding optimum temperature and light intensity was observed for TKS as well as for IWT: the optimum temperature for males was higher than for females (+0.8 °C); and the optimum light intensity for males was lower than for females (−5.3 µmol m^−2^ s^−1^). For KGS, the optimum temperature indicated was 20.6 °C for both males and females. The optimum light intensity for males was higher than for females (+17.5 µmol m^−2^ s^−1^). The maturity for KGS progressed under white light for both males and females. Comparing regional lines, the optimum temperature for males was the same for both IWT and TKS, and for KGS was 1.1 °C higher than the others. The optimum temperature for females increased in the order IWT (17.9 °C), TKS (18.7 °C), KGS (20.6 °C). The optimum light intensity for males increased in the order IWT (39.8 µmol m^−2^ s^−1^), TKS (39.3 µmol m^−2^ s^−1^), KGS (50 µmol m^−2^ s^−1^). In contrast, the optimum light intensity for females decreased in the order IWT (50 µmol m^−2^ s^−1^), TKS (44.6 µmol m^−2^ s^−1^), KGS (32.5 µmol m^−2^ s^−1^).

The predicted changes in maturation of male and female gametophytes for the three regional lines is shown in Figure 5, assuming cultivation under the optimal conditions obtained from the models (Table 6). For IWT, the combined proportion of maturation stages IV and V of male gametophytes reached over 50% (IV: 48.1%, V: 9.7%) on day 13, and achieved approximately 100% on day 19 (IV: 19.5%, V: 78.4%). The progress of female maturation was slower, with maturity stages IV and V together at about 30% on day 13, and 69% on day 19. The difference in maturation speed between males and females showed a similar trend among the three-regional lines (Figure 5). We have partly confirmed that the degree of maturation progresses according to the predicted changes. In the cultivation study with assays No. 2 and No. 6 (Table 2), which are similar to optimum conditions, changes of maturation coincided more or less with the prediction for both males and females. 

## 4. Discussion

The experimental results (Table 1 and Table 2) and the optimum conditions predicted by RSM (Table 4 and Table 6) revealed differences in optimum temperature and light intensity for growth of male versus female *U. pinnatifida* gametophytes: optimum temperature and light intensity were both higher for males than for females. Moreover, maturity progressed faster in males than females (prediction of Figure 5 based on data of Table 2). These characteristics were common to all three regional lines, thus there is a clear physiological sexual dimorphism in the ecological responses of *U. pinnatifida* gametophytes. 

Sex-related differences in gametophyte response to temperature have been discussed for another kelp, *Saccharina latissima*, in which a larger number of male gametophytes than female gametophytes was found (in the open sea) at higher temperatures [45]. At a high temperature, 20 °C, transcripts of female *S. latissima* gametophytes, but not males, changed drastically [46], which has been interpreted as signifying that males have a higher thermal tolerance than females [47], and was consistent with Norton’s results [48]. Our results suggest that in *U. pinnatifida*, the female gametophytes appear to be more sensitive to elevated temperatures than the male gametophytes. Importantly, in the present study, there were no differences among the optima for temperature and light intensity affecting male gametophyte growth among the three-regional lines, implying that the sexual difference appears to be a consequence of natural selection of female gametophytes in the natural coastal environment. In addition, the optimum temperature range among the regional lines in growth and maturation of female gametophytes was larger than for that of males. This appears to be best explained by the greater sensitivity of female gametophytes to higher temperatures. This restriction on female maturation may contribute to the absence of sporophyte production at high temperatures, because sporophytes of laminarian kelps are more sensitive to high temperatures than their male and female gametophytes [49,50,51]. 

In the present study, the optimum environmental conditions for growth and maturation of the gametophytes were also different among regional lines. Considering growth, among the three regional lines, the optimum temperature was highest and light intensity was lowest in KGS gametophytes. Considering maturation, the optimum temperature and light intensity in IWT and TKS gametophytes were about the same, with higher optimum temperature and lower optimum light intensity for males than for females. However, in KGS gametophytes, the optimum temperature for growth was the same for males and females, but the optimum light intensity for females was higher than that for males. According to information on haplotype divergence of the mitochondrial loci of *U. pinnatifida*, the regional lines including IWT and TKS have the same phylogenetic features and belong to a different group than KGS [44]. Therefore, the difference of growth and maturation in male and female gametophytes between the ITW/TKS group and KGS may be due to genetic divergence, although we cannot exclude the possibility of environmental effects, such as epigenetic responses, occurring because of temperature difference at sampling: mean temperatures were different among the three sampling sites. The coast of Kagoshima Pref., where the KGS mother plant was collected, is located at the southern limit of the distribution of *U. pinnatifida* in Japan [27]. The coastal seawater temperature in summer often exceeds 30 °C (Kagoshima Pref.), which is above the growth-limiting temperature of this species (28 °C, [23]). The optimum light intensity for the female KGS gametophytes was the lowest among the three regional lines, suggesting that KGS *U. pinnatifida* is better adapted to growth in deeper water and thus avoids damage by elevated temperature and light intensity near the surface. Furthermore, in male and female gametophytes from KGS, no apparent interaction effect was detected between temperature and light intensity for either growth or maturation (Appendix A). Therefore, to accommodate their more widely fluctuating natural environmental conditions, KGS gametophytes may be able to grow and mature more rapidly under conditions where one or other of either temperature or light intensity is optimal.

In previous research on gametophytes, the optimum temperature for growth and upper temperature limit for reproduction were reported for nine laminarian species on the coast of California revealing temperature differences between central and southern California of 5 °C for growth and 3 °C for reproduction [52]. Morita et al. (2003) reported that the optimum temperature difference for maturation between *U. pinnatifida* and *U. undarioides* was 5 °C, concluding that the difference is a major factor determining the distribution differences of these species [23]. Both studies indicated species distribution differences affected by optimal temperatures for growth and maturation during the gametophyte stage. In addition, several morphological and physiological ecotypes have been observed in the sporophyte stage of *U. pinnatifida* [28,29,30]. The present results revealed that this alga also shows not only sexual differences but also ecological differences in physiological characteristics at its gametophytic stage: the optimum temperature for maturation in KGS was higher than in IWT and TKS. Furthermore, although the optimum temperature for growth in male gametophytes showed almost no difference among the three regional lines, the temperature optimum for female gametophytes varied by 3.3 °C. Similarly, the regional variation in optimum temperature for maturation of female gametophytes was 2.5 °C compared to only 0.9 °C for males. Therefore, regional differences in ecotypical characteristics appear to be larger in females than in males. This flexibility of *Undaria* gametophytes in accepting a range of environmental conditions may help to explain why this species has successfully established itself globally within a short period of time.

The response of the gametophytes to wavelength was identical among the sexes and across all three regional lines. Blue (400–500 nm) light stimulated the maturation of gametophytes as described previously for *U. pinnatifida* female gametophytes [21,24,53]. Green (500–600 nm) light stimulated growth of gametophytes without maturation, while the presence of only red (600–700 nm) light has a negative effect on growth and maturation. In other members of the Laminariales, egg and sporophyte formation under blue light and their delayed formation under red light have been reported for female gametophytes of *Laminaria saccharina* [54,55]; and activation of the synthesis of photosynthetic pigments and carbon synthesis under the influence of blue light have been revealed in *Saccharina japonica* through transcriptome analysis [56]. These responses to blue light may be related to the blue-light receptor aureochrome [57]. 

There are a few reports of responses under green light in brown algae, although very little gamete release under the influence of light was found in *Silvetia compressa* following experiments across a light wavelength range from green to red [58]. From the results of the present study, it is difficult to judge whether the growth promotion seen under green light is the effect of green light itself or a result of the absence of blue light. However, the responses of *U. pinnatifida* gametophytes under green and blue light would allow the development of an industrial sporeling method with higher synchronicity in which gametophytes can be grown without maturation under green light, and then mature under blue light. Gametophyte growth under white light was lower than under blue or green, suggesting that the biological response of gametophytes depends on the proportion of individual wavelengths available. White light sources are generally used for cultivation experiments of algae and almost all light sources have been LED, so it is clear that more detailed wavelength information about light sources is necessary to ensure reproducibility.

An additional effect of blue light was to stimulate the progress of parthenosporophytes on female gametophytes (i.e., sporophyte formation without fertilization). Such parthenogenetic development has been observed in *U. pinnatifida* [59] and several other species of the Laminariales [60,61]. Although the seasonal frequency of parthenogenesis in female gametophytes of *Laminaria nigrescens* has been observed with a maximum in spring to early summer [61], the trigger for parthenogenesis requires further investigation. This blue-light response may be a key factor for the progress of parthenogenesis of female gametophytes in the Laminariales.

Previously, studies of environmental factors for growth and maturation in *U. pinnatifida* gametophytes and sporophytes were focused on searching for the optimum values of each single environmental factor and the threshold value in order to discuss horizontal or vertical distribution limits. However, in discussing adaptations to environmental changes among sexes and regional lines, a single-factor analysis between environmental factors and the biological response is inadequate because growth and maturation of macroalgae are influenced by a complex interaction of various environmental factors including temperature, light intensity, wave action, and nutrient concentration [62,63]. The experimental design and analysis methods used in the present study are powerful tools for understanding the optimum environmental conditions, and their interactions, for macroalgal growth and maturation. This methodology can be adopted as an effective way to screen natural populations to obtain elite cultivars. 

Recently, in order to improve the cultivation yield of *U. pinnatifida* at localities exposed to higher environmental temperatures, cross-breeding has been used in an attempt to establish a high-temperature-resistant line by using male and female gametophytes collected at various regional localities [64]. The experimental design and RSM analysis method, in conjunction with cross-breeding, may accelerate the achievement of success in increasing the commercial yield of *U. pinnatifida*.

## 5. Conclusions

Male and female gametophytes of *Undaria pinnatifida* revealed differences in temperature and light intensity optima for growth and maturation in the cultivation experiments and RSM analysis. Female gametophytes appear to be more susceptible to temperature than males. Since these characteristics were found to be common to three regional lines, our data provide a clear evidence for physiological sexual dimorphism in the ecological responses of *U. pinnatifida* gametophytes. The present study has also demonstrated the existence of ecotypic differences at the gametophyte stage, considering that the optimum conditions for growth and maturation were different among the three regional lines. In contrast, all three regional lines indicated common responses to light wavelength: blue light stimulated maturation, green light stimulated growth without maturation, and under red light there was very little vegetative growth. These results provide useful information allowing the development of an industrial sporeling method.

## Figures and Tables

**Figure 1 genes-11-00944-f001:**
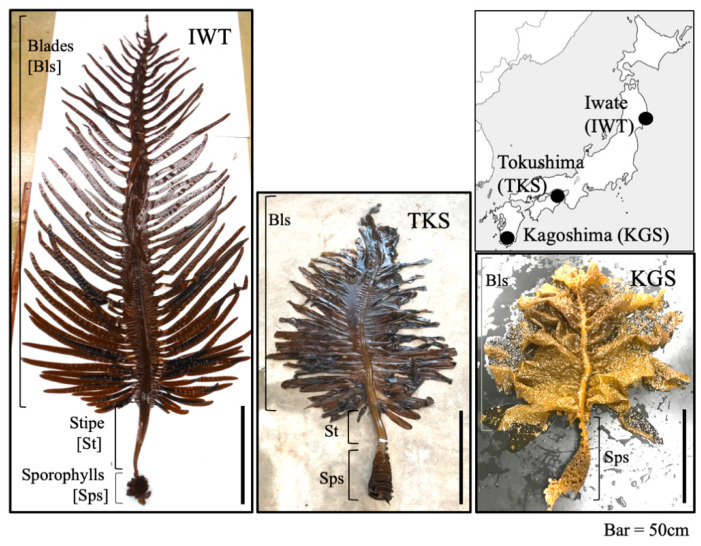
The three localities where *Undaria pinnatifida* thalli were collected and the corresponding sporophytes. Scale bar 50 cm. IWT—Iwate, TKS—Tokushima, KGS—Kagoshima.

**Figure 2 genes-11-00944-f002:**
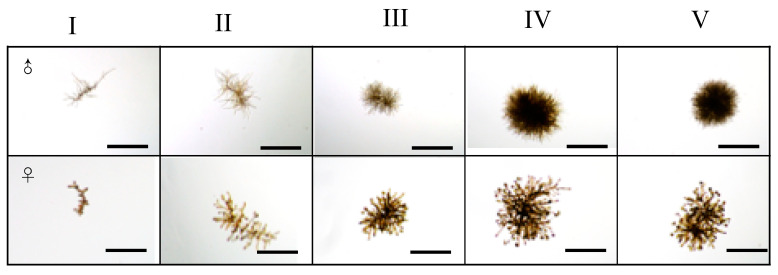
Illustrating the maturity scales for male and female gametophytes, from I (immature) to V (fully mature). Scale bars 500 µm.

**Figure 3 genes-11-00944-f003:**
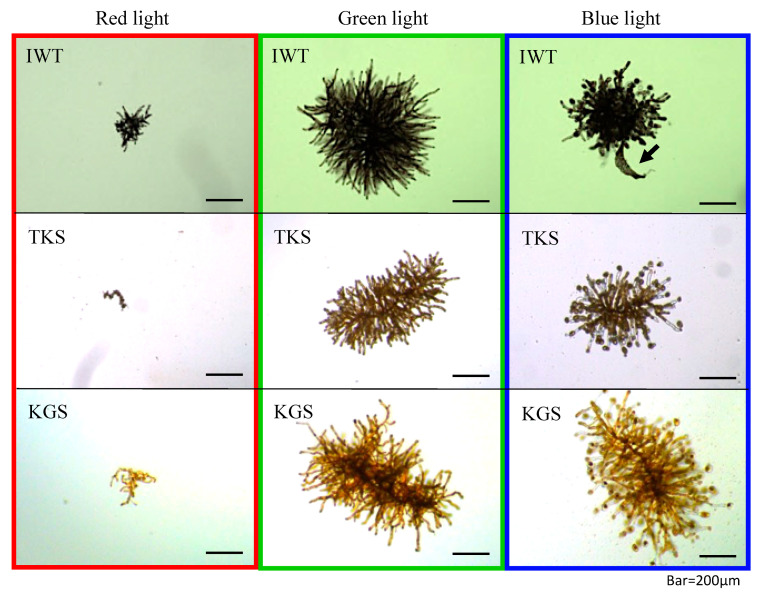
Photographs of representative female gametophytes from the three regions (IWT—Iwate, TKS—Tokushima, KGS—Kagoshima), cultured for 25 days under different wavelengths of light. As a representative condition, these gametophytes were cultivated at 40 µmol m^−2^ s^−1^ and 20 °C under each light color. Scale bars 200 µm. The arrow on the IWT image in blue light indicates sporophyte formation.

**Figure 4 genes-11-00944-f004:**
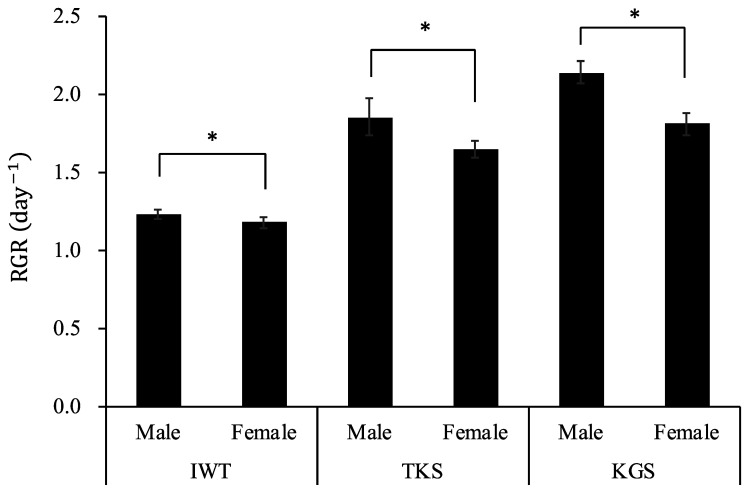
Comparison of relative growth rates (RGR) of male and female gametophytes of *Undaria pinnatifida* for the three regional lines cultivated under optimum temperature, light intensity, and wavelength. Values are expressed as mean ± SE (*n* = 24). * indicates significant differences among the means of males and females (*t*-test, *p* < 0.05). IWT—Iwate, TKS—Tokushima, KGS—Kagoshima.

**Figure 5 genes-11-00944-f005:**
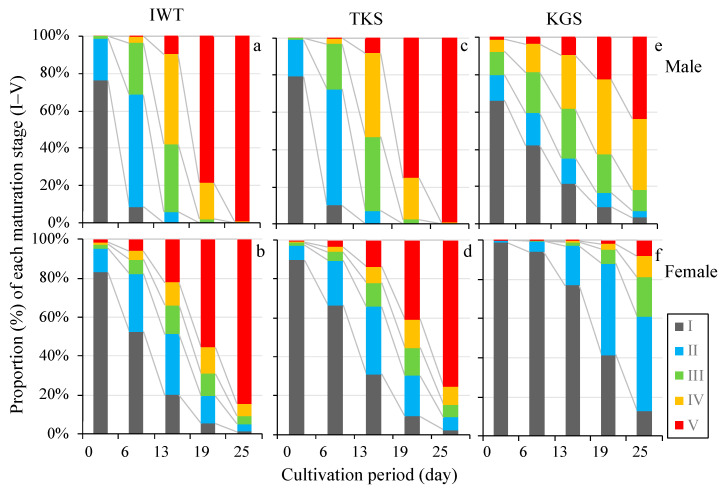
Predicted changes in the maturation stage of male and female gametophytes for the three regional lines, under the optimal conditions obtained from the models (Table 6) at each cultivation day (0, 6, 13, 19, 25): IWT (**a**, male; **b**, female), TKS (**c**, male; **d**, female), and KGS (**e**, male; **f**, female). The stages correspond to those in Figure 2. IWT—Iwate, TKS—Tokushima, KGS—Kagoshima.

**Table 1 genes-11-00944-t001:** Results of growth experiments for male and female gametophytes of *Undaria pinnatifida* and the observed response variables (RGR, relative growth rate day^−1^). The variables matrix was decided following the design protocol (see Materials and Methods) IWT—Iwate, TKS—Tokushima, KGS—Kagoshima.

Assay	Factor 1 (X_1_)	Factor 2 (X_2_)	Factor 3 (X_3_)	Response (Y, RGR)						
	Temperature (°C)	Light Intensity (µmol m^−2^ s^−2^)	Light Color	Male			Female			
				IWT	TKS	KGS		IWT	TKS	KGS
1	15	2	Green	0.504	0.780	1.073		0.538	0.714	1.022
2	15	21	Red	0.473	0.627	1.114		0.649	1.006	1.090
3	15	21	Blue	0.960	1.250	1.905		1.018	1.522	1.587
4	15	40	Green	1.09	1.459	1.724		1.042	1.573	1.515
5	16	32	White	1.109	1.293	1.887		1.066	1.494	1.524
6	20	2	Blue	0.771	1.017	1.343		0.689	0.783	1.043
7	20	2	White	0.739	0.841	1.218		0.617	0.627	0.932
8	20	21	Green	1.231	1.743	1.971		1.065	1.527	1.772
9	20	40	Blue	1.277	1.672	1.994		1.134	1.517	1.536
10	25	2	Red	0.224	0.246	0.454		0.128	0.193	0.413
11	25	2	Green	0.733	1.004	1.143		0.460	0.709	0.992
12	25	21	Blue	1.110	1.450	1.975		0.881	1.393	1.556
13	25	40	Red	0.652	0.878	1.105		0.515	0.804	0.907
14	25	40	White	1.107	1.325	1.770		0.812	1.165	1.292
15	25	40	Green	1.161	1.635	1.794		0.847	1.32	1.485

**Table 2 genes-11-00944-t002:** Results of experiments to observe the effects of environmental parameters on maturation (stages I to V) of male (upper matrix) and female (lower matrix) gametophytes of *Undaria pinnatifida.* Observations: probability of maturation for each group on cultivation day 25.

Sex	Assay	Factor 1 (X_1_)	Factor 2 (X_2_)	Factor 3 (X_3_)	Maturation Degree at the Final Date of Cultivation														
		Temperature (°C)	Light Intensity(µmol m^−2^ s^−2^)	Light Color	IWT					TKS					KGS				
					1	2	3	4	5	1	2	3	4	5	1	2	3	4	5
	1	10	10	Blue	0.00	0.00	0.00	0.00	1.00	0.00	0.00	0.03	0.20	0.77	0.91	0.04	0.03	0.01	0.00
	2	10	30	White	0.00	0.00	0.00	0.02	0.98	0.00	0.00	0.04	0.22	0.74	0.83	0.08	0.06	0.03	0.01
	3	10	50	Blue	0.00	0.00	0.01	0.33	0.66	0.00	0.00	0.02	0.15	0.82	0.54	0.16	0.17	0.10	0.02
	4	16	26	White	0.00	0.00	0.00	0.02	0.98	0.00	0.00	0.00	0.02	0.98	0.00	0.00	0.00	0.02	0.98
	5	16	26	White	0.00	0.00	0.03	0.51	0.46	0.00	0.00	0.00	0.02	0.98	0.00	0.00	0.00	0.02	0.98
Male	6	17	50	White	0.00	0.00	0.00	0.02	0.98	0.00	0.00	0.00	0.01	0.98	0.04	0.03	0.11	0.38	0.43
	7	18	10	Blue	0.00	0.00	0.00	0.09	0.90	0.00	0.00	0.00	0.01	0.99	0.11	0.09	0.23	0.38	0.19
	8	19	34	Blue	0.00	0.00	0.02	0.33	0.65	0.00	0.00	0.00	0.01	0.99	0.11	0.09	0.22	0.39	0.20
	9	19	34	Blue	0.00	0.00	0.00	0.00	1.00	0.00	0.00	0.00	0.01	0.99	0.06	0.05	0.16	0.41	0.32
	10	25	10	White	0.00	0.00	0.00	0.06	0.94	0.00	0.00	0.00	0.02	0.98	0.08	0.07	0.20	0.41	0.24
	11	25	30	Blue	0.00	0.00	0.00	0.02	0.98	0.00	0.00	0.00	0.01	0.99	0.00	0.00	0.00	0.01	0.99
	12	25	50	White	0.00	0.00	0.01	0.22	0.77	0.00	0.00	0.00	0.01	0.98	0.10	0.08	0.21	0.40	0.22
	1	10	10	Blue	0.84	0.12	0.02	0.01	0.01	0.94	0.05	0.01	0.00	0.01	1.00	0.00	0.00	0.00	0.00
	2	10	30	White	0.80	0.15	0.02	0.01	0.01	0.86	0.10	0.02	0.01	0.01	1.00	0.00	0.00	0.00	0.00
	3	10	50	Blue	0.30	0.37	0.13	0.09	0.12	0.82	0.13	0.02	0.01	0.02	1.00	0.00	0.00	0.00	0.00
	4	16	26	White	0.13	0.28	0.16	0.15	0.28	0.26	0.34	0.14	0.09	0.18	0.56	0.37	0.04	0.02	0.01
	5	16	26	White	0.13	0.28	0.16	0.15	0.28	0.26	0.34	0.14	0.09	0.18	0.56	0.37	0.04	0.02	0.01
	6	17	50	White	0.04	0.12	0.11	0.15	0.58	0.12	0.24	0.15	0.13	0.36	0.50	0.41	0.05	0.02	0.01
Female	7	18	10	Blue	0.10	0.25	0.16	0.16	0.34	0.28	0.34	0.13	0.08	0.16	0.95	0.05	0.00	0.00	0.00
	8	19	34	Blue	0.02	0.06	0.06	0.10	0.76	0.03	0.07	0.07	0.09	0.74	0.19	0.52	0.16	0.08	0.05
	9	19	34	Blue	0.02	0.06	0.06	0.10	0.76	0.03	0.07	0.07	0.09	0.74	0.19	0.52	0.16	0.08	0.05
	10	25	10	White	0.91	0.07	0.01	0.01	0.01	0.78	0.16	0.03	0.01	0.02	0.89	0.10	0.01	0.00	0.00
	11	25	30	Blue	0.30	0.37	0.13	0.09	0.12	0.07	0.17	0.13	0.13	0.50	0.81	0.17	0.01	0.01	0.00
	12	25	50	White	0.44	0.35	0.09	0.05	0.07	0.18	0.30	0.15	0.11	0.25	0.87	0.12	0.01	0.00	0.00

**Table 3 genes-11-00944-t003:** ANOVA table of refined models for male and female gametophytes of *Undaria pinnatifida*.

Line	Sex	Source	Sum of Square	*df*	Mean Square	*F*	Significance
IWT	Male	Regression	31.576	8	3.947	179.95	<0.0001
		Residual	6.953	317	0.022		
		Total	38.529	325			
	Female	Regression	25.278	8	3.160	174.42	<0.0001
		Residual	5.743	317	0.018		
		Total	31.021	325			
TKS	Male	Regression	54.291	8	6.786	61.52	<0.0001
		Residual	34.305	311	0.110		
		Total	88.596	319			
	Female	Regression	57.178	8	7.147	153.79	<0.0001
		Residual	14.500	312	0.046		
		Total	71.678	320			
KGS	Male	Regression	56.814	7	8.116	149.60	<0.0001
		Residual	14.431	266	0.054		
		Total	71.245	273			
	Female	Regression	44.490	7	6.356	163.47	<0.0001
		Residual	13.530	348	0.039		
		Total	58.019	355			

IWT—Iwate, TKS—Tokushima, KGS—Kagoshima.

**Table 4 genes-11-00944-t004:** The temperature, light intensity, and light wavelength for optimal growth of male and female gametophytes for Iwate (IWT), Tokushima (TKS), and Kagoshima (KGS) lines, as predicted by response surface methodology.

Environmental Factor	IWT		TKS		KGS	
	Male	Female	Male	Female	Male	Female
Temperature (°C)	20.7	18.6	20.9	16.5	20.7	19.8
Light intensity (µmol m^−2^ s^−1^)	33.7	32.5	32.7	31.3	28.6	26.9
Light color	Blue	Blue	Green	Blue	Blue	Blue

IWT—Iwate, TKS—Tokushima, KGS—Kagoshima.

**Table 5 genes-11-00944-t005:** Likelihood ratio tests using a refined ordered logistic regression model for maturation of male and female gametophytes at Iwate (IWT), Tokushima (TKS), and Kagoshima (KGS).

Line	Sex	Log Likelihood × (−1)	*df*	*𝝌* *2*	Significance
IWT	Male	1475.97	7	3827.07	<0.001
	Female	1496.28	7	1461.8	<0.001
TKS	Male	1549.45	7	3625.8	<0.001
	Female	1148.54	7	1240.27	<0.001
KGS	Male	226.21	7	133.92	<0.001
	Female	461.88	7	552.48	<0.001

IWT—Iwate, TKS—Tokushima, KGS—Kagoshima.

**Table 6 genes-11-00944-t006:** The optimized temperature, light intensity, and light color for maturation of male and female gametophytes in the Iwate (IWT), Tokushima (TKS), and Kagoshima (KGS) lines, as predicted by ordered logistic regression.

Environmental Factor	IWT		TKS		KGS	
	Male	Female	Male	Female	Male	Female
Temperature (°C)	19.5	17.9	19.5	18.7	20.6	20.6
Light intensity (µmol m^−2^ s^−1^)	39.8	50	39.3	44.6	50	32.5
Light color	Blue	Blue	Blue	Blue	White	White

IWT—Iwate, TKS—Tokushima, KGS—Kagoshima.

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
