# Peer review of "Sexual Difference in the Optimum Environmental Conditions for Growth and Maturation of the Brown Alga Undaria pinnatifida in the Gametophyte Stage"

_genes, 2020, doi:10.3390/genes11080944_

Round 1
Reviewer 1 Report
Please attached comments and suggestions for authors

Reviewer 2 Report
The presented article presents an up-to-date research of fundamental importance and potentially applicable in aquaculture.Annual brown kelp - Undaria pinnatifida wich is an growing naturally in coastal areas as a major primary producer in temperate regions.
In general, the article is written clearly and can be printed after correcting a number of technical errors.
line 26 - apparently it is required to delete "453 nm" (500-600 nm; λmax 525 nm 453 nm)
line 149 - There is no bar ruler in the photo of figure 2
line 180 - The table looks untidy; for improvement, add 0 as the second decimal place and align the data to the right bord.
line 184 - The table looks untidy; for improvement, add 0 as the second decimal place and align the data to the right bord.
line 310 - Add a number after the word picture.
line 311 - Add the stage names to the figure caption, as figures should contain complete information.
line 435 - The conclusion in the article is not clearly expressed. Probably, the clear result should be emphasized and the regulation of which parameter, according to the authors, should be specified, and the range in which this conclusion is valid should be indicated.
I think that this article will be minor revise.
